# Fungal Innovations—Advancing Sustainable Materials, Genetics, and Applications for Industry

**DOI:** 10.3390/jof11100721

**Published:** 2025-10-06

**Authors:** Hannes Hinneburg, Shanna Gu, Gita Naseri

**Affiliations:** 1Fraunhofer Institute for Applied Polymer Research, Geiselbergstrasse 69, 14476 Potsdam, Germany; 2Institut für Biologie, Humboldt-Universität zu Berlin, Philippstrasse 13, 10115 Berlin, Germany; shanna.gu@hu-berlin.de; 3Max Planck Unit for the Science of Pathogens, Charitéplatz 1, 10117 Berlin, Germany

**Keywords:** bioprocessing, combinatorial optimization, fungal material, synthetic biology, sustainability

## Abstract

Fungi play a crucial yet often unnoticed role in our lives and the health of our planet by breaking down organic matter through their diverse enzymes or eliminating environmental contamination, enhancing biomass pretreatment, and facilitating biofuel production. They offer transformative possibilities not only for improving the production of materials they naturally produce, but also for the production of non-native and even new-to-nature materials. However, despite these promising applications, the full potential of fungi remains untapped mainly due to limitations in our ability to control and optimize their complex biological systems. This review focuses on developments that address these challenges, with specific emphasis on fungal-derived rigid and flexible materials. To achieve this goal, the application of synthetic biology tools—such as programmable regulators, CRISPR-based genome editing, and combinatorial pathway optimization—in engineering fungal strains is highlighted, and how external environmental parameters can be tuned to influence material properties is discussed. This review positions filamentous fungi as promising platforms for sustainable bio-based technologies, contributing to a more sustainable future across various sectors.

## 1. Introduction

Fungi are an integral part of our daily lives, often unnoticed yet profoundly impactful. These remarkable organisms play a crucial role in various sectors, from environmental conservation to sustainable industries [1,2,3,4,5]. Through their diverse enzymatic capabilities, fungi break down complex organic matter, aiding in waste decomposition and the elimination of environmental pollutants [4,6]. They produce valuable metabolites such as antibiotics [7], vitamins [5], and enzymes used in medicine [5], agriculture [5], and industrial processes, including paper production and food processing [1,3,5]. Furthermore, fungi enhance biomass pretreatment, facilitating the production of biofuels and reducing reliance on fossil fuels [1,4]. The innovative use of fungal-based materials is revolutionizing sustainable construction [8], textiles [9,10], packaging [1,11], and composite materials [1,11,12].

Mycelium—a network-like fungal structure—serves as the ecophysiological unit that allows filamentous fungi to colonize substrates, fulfill ecological functions, sense the environment, and absorb nutrients that have been previously digested with secreted hydrolytic and oxidative enzymes, ultimately playing a vital role in nutrient recycling and provision within ecosystems [13,14,15].

While mycorrhizal fungi exhibit filamentous growth and form symbiotic relationships with plant roots, enhancing nutrient uptake and improving soil health [16], they are not the focus of this review. However, it is worth mentioning that their role in agriculture is critical, as they promote sustainable farming practices by increasing crop resilience and reducing the need for chemical fertilizers. In addition, they contribute to bioremediation efforts that help break down pollutants in contaminated soils [6], highlighting their overall importance in ecosystem health and sustainability. These contributions underscore the vital role of fungi in addressing global challenges and promoting sustainable solutions.

Saprotrophic fungi contribute to the decomposition of wood and other plant materials, facilitating the recycling of nutrients back into the soil [17]. This specificity is important, as filamentous fungi, from the phylum Basidiomycota, class Agaricomycetes, including genera such as *Pleurotus*, *Ganoderma*, *Fomes*, and *Trametes*, are the primarily studied organisms for composite formulations and are central to the production of mycelium-based composites. The majority of fungal species used for the cultivation of biomaterials have been reviewed in the past [12,13,18] with a focus on these same genera, along with others such as *Agaricus*, *Polyporus*, and *Pycnoporus*. Several Basidiomycota filamentous fungi present saprophytic lifestyles on remains of dead plant biomass, but there are also facultative phytopathogens of different tree species [15,19].

The mycelium [14] itself is a complex, dynamic network of elongated cells, called hyphae (Figure 1). Hyphae grow as filaments through their surroundings, literally fusing with the materials around them, leading to the emergence of composite materials that can be deliberately influenced during growth. The adaptability of filamentous fungi enables the creation of new composite materials with various textures and densities, opening up potential applications in construction, textiles, fashion, biomedical fields, and packaging, using fungal networks directly or as biological binders [11,18]. The use of pure mycelium for various material applications is also documented, often aiming to produce more flexible materials [9]. Furthermore, fruiting bodies from the wild, such as those from bracket fungi like the tinder fungus *Fomes fomentarius*, are processed by experienced craftsmen in certain areas of Europe into accessories, including hats, bags, jewelry, and similar items [20,21].

The filamentous, wood-decaying fungi have a cell wall, which is primarily composed of β-glucans and chitin, both strong and flexible polysaccharides, as well as several proteins, such as mannoproteins [13,22,23]. Chitin contributes to cell wall rigidity, while β-glucans enhance flexibility and structural integrity. Variations in the composition of these components can influence the mechanical properties of the resulting mycelium composites [24,25,26]. Their remarkable ability to live off lignocellulosic biomass, along with their unique material properties, positions filamentous fungal materials as a valuable resource for developing sustainable, biodegradable materials. These materials can be used, among other things, to create leather alternatives [9,10], insulation [27], building [27], or regenerative biomaterials [28] and packaging [11,29]. A key advantage of mycelium-based materials from filamentous fungi is their adaptability. By selectively choosing the type of fungus, the cellulose-rich waste materials, and the growing conditions, as well as suitable post-growth processing methods, properties such as strength, elasticity, and density can be tailored to meet specific needs within biological boundaries. This allows for customized solutions for various applications [13,18]. These materials are not only fairly robust and versatile but also possess heat-insulating, moisture-regulating, and fire-resistant qualities, making them particularly attractive for use in constructing certain applications [18].

It is crucial to note that the mechanical properties of a fungal composite are more influenced by the chosen species, rather than the specific strain. For example, materials derived from *Trametes versicolor* exhibit greater compressive strength and stiffness compared to those made from *Pleurotus ostreatus*. Additionally, *P. ostreatus*-based composites demonstrate stiffness double that of those derived from *Ganoderma lucidum*, with measured values of 28 MPa and 12 MPa, respectively. These differences can likely be attributed to the higher content of polysaccharides present in composites made from *P. ostreatus* [13,18]. However, strain-related differences within a genus also exist, for example, when strains have specific growth rates on a given substrate/growth regime, ultimately impacting the material properties [30]. As we explore fungal innovations, we uncover the potential of fungal materials and genetic advancements leading to sustainable solutions [31,32]. However, as the unique potential of filamentous fungi remains largely underexplored due to a lack of sufficient genetic engineering strategies, this review aims to synthesize key developments in synthetic biology and fabrication approaches, to illustrate emerging opportunities for the potential of filamentous fungi for sustainable and biodegradable material applications.

## 2. Fungal Materials

Fungal materials based on filamentous species are versatile and sustainable, offering both rigid and flexible applications. Rigid fungal composites can replace traditional building materials [18] and packaging, while flexible options are ideal for innovations in textiles [9]. This section explores the properties and potential uses of these diverse fungal materials in promoting eco-friendly solutions.

### 2.1. Rigid Fungal Materials

Rigid fungal composites are created by combining fungi with lignocellulosic fibers or particles, producing materials with varying properties depending on the used finishing method (e.g., hot/cold pressing) [29,33], followed by the used substrate [27,33], as well as fungal species and strains (whereas species influences are more pronounced) [18], particularly the growth behavior [30,33] and hyphal type [33] (monomitic, dimitic, or trimitic, see Figure 1), besides substrate nutritional profile and growth conditions [29,33]. For example, *Ganoderma* has a trimitic system composed of generative, skeletal, and ligative (binding) hyphae. The alignment of skeletal hyphae provides additional mechanical resistance to loading in specific directions, thereby enhancing the overall strength of the composite [34].

The biological nature of the materials leads to performance fluctuations while also ensuring a high degree of biodegradability [18,25,27,29]. These composites exhibit common mechanical and physical characteristics [12,25,27,29,33], summarized in Table 1. A comprehensive overview of their material characteristics can be found in articles by Sharma et al. [12], Aiduang et al. [18], and Jones et al. [27]. While their mechanical strength and moisture uptake limit their use primarily to non-weight-bearing applications, such as interior panels and acoustic absorption [29], these materials are biodegradable and have demonstrated potential in architectural designs [8,35]. Notable examples include the following:Mycotectural Alpha (2009): Utilized *G. lucidum*-bound sawdust for its construction.Hy-Fi (2014): A cluster of circular towers made from mycelium-based bricks.MycoTree (2017): Featured mycelium-bound composite blocks in its installation.MycoTree 2.0 (2019): An expanded version that continued the use of mycelium-bound composite blocks.Growing Pavilion (2020): Incorporated *Ganoderma lingzhi* mycelium composite panels mounted on wooden frames.My-Co Space (2021): Showcased elements of hemp-grown *F. fomentarius* on a supporting structure.

More recently, research is exploring the use of extrudable mycelium-based composite pastes for additive manufacturing (AM) and 3D-printed structures [36,37,38,39]. Together, these studies highlight the potential of rigid materials to contribute to eco-friendly innovation while addressing both functional and visual needs.

### 2.2. Flexible Fungal Materials

Flexible fungal materials have diverse applications, including fungal wound dressings [40] (e.g., *F*. *fomentarius*), medical cell scaffolds [28], paper-like materials [41,42], fungal chitin nanomaterials [43,44,45], filters for water treatment [45,46], and meat analogs [47]. Fungal materials derived from fruiting bodies (using the inner fibrous tissue or ‘Trama’) of *F. fomentarius* and *Fomitopsis betulina* (syn *Piptoporus betulinus*) have been transformed into soft goods such as hats, bags, tablecloths, and scarves for centuries, with this practice still ongoing in parts of Romania [20] and other regions of Europe, such as Germany [21]. However, due to their limited availability and fragility, biotechnologically produced alternatives are being explored, such as foam and leather-like materials [9,10,11]. Lab-grown materials, pure or as composites, leverage the ability of hyphae to interweave and connect, forming a mycelium network within days or weeks. These materials are sustainable, biodegradable, and customizable, and their properties depend on the fungal strain, substrate, growth regime, and post-processing techniques (e.g., drying, pigmenting, plasticizing) for enhancing (microbiological) robustness and appearance [9,10,11]. Mycelium-based foams and leather alternatives, made from agricultural waste, are cruelty-free and more eco-friendly than traditional materials, as they generate less pollution and use less water. They are lightweight, offer good thermal and acoustic insulation, and require fewer resources to produce [9,10,48]. Several genera, such as *Ganoderma*, *Trametes*, and *Fomes*, are utilized for leather-like mycelium materials, although not all produce commercially viable materials [9,48,49] mainly due to a lack of information, especially regarding patents [49].

Properties such as density, tensile strength, and elasticity vary depending on the fungal strain, growth conditions, and post-processing methods. Leather-like mycelium materials typically have densities ranging from 240 to 800 kg/m^3^, tensile strengths from 5.6 to 14 MPa, and elastic moduli from 13 to 55 MPa, making them less robust than bovine leather [9]. The mechanical properties reported in direct comparisons between genera under the same conditions are usually lower than those in commercial production, likely due to various factors, including a lack of knowledge of post-processing methods and different growth conditions. For instance, Kniep et al. [50] reported tensile strengths of *F. fomentarius*, *Pleurotus eryngii*, and *T. versicolor* to be approximately 3.3 MPa, 5.3 MPa, and 4.2 MPa, respectively. Foams are less robust, with densities ranging from 30 to 50 kg/m^3^ and tensile strengths from 0.1 to 0.3 MPa [9,11].

Approximately 20 companies lead the mycelium product business [48]. Notable companies, including Ecovative, which creates biodegradable packaging; ForagerBio (a brand under Ecovative), which produces mycelium materials/foams for fashion; MOGU, which offers mycelium-based products, including leather lookalikes; MycoWorks, which provides fine leather alternatives; Mycotech Lab, which produces mycelium ‘leather’; and COMU Labs, which makes mycelium-based furniture.

## 3. Synthetic Biology Toolbox for Genetic Engineering of Fungal Materials

Synthetic biology has revolutionized metabolic engineering by providing new tools and methodologies for creating and engineering synthetic genetic circuits in microbial cell factories. The invented tools and methods facilitate the expression of genes of biosynthetic pathways or biosynthetic gene clusters (BGCs), enabling microbial cell factories to produce high-value secondary metabolites. Model organisms *Escherichia coli* and *Saccharomyces cerevisiae* serve as standard hosts for synthetic biology due to their rapid growth, simple genetics, and well-established engineering toolkits. Nevertheless, their limitations in RNA splicing and complex regulation highlight why filamentous fungi are essential for advancing fungal materials research [51]. Filamentous fungi serve as powerful biomanufacturing platforms due to their exceptional structural and biosynthetic capabilities, as well as their remarkable adaptability. Using either classical or synthetic biology-assisted metabolic engineering approaches, various proteins and metabolites have been produced in filamentous fungi, as summarized in Liu et al. [31]. To further unlock the potential of mycelium-based materials, this review discusses three key and powerful toolsets that their tailored implementation in filamentous fungi can drive significant breakthroughs in fungal material engineering: transcriptional regulation tools, genome editing techniques, and rapid DNA assembly methods. Notably, the first two—transcriptional control and genome editing—serve as tools that enable the third, intending to shorten the design-build-test-learn cycle of engineering mycelium fungi for material production.

### 3.1. Regulation Reprogramming Tools

Regulation reprogramming tools are vital for harnessing filamentous fungi in synthetic biology. They enable precise gene expression and metabolic control, optimizing fungi for the production of bio-based materials, proteins, and metabolites. To successfully express a gene in a heterologous host, a transcriptional unit consisting of promoters and terminators is needed to, respectively, initiate and terminate transcription of the gene of interest [52]. While terminators for filamentous fungi have been comprehensively reviewed by Fan et al. [32]. This section focuses on the application of synthetic transcription factors (synTFs) as a promising alternative to promoters for transcriptional control of gene expression in filamentous fungi.

In filamentous fungi, native promoters such as PgpdA [53] and other carbon- and nitrogen-dependent promoters [54,55,56,57,58,59] are often utilized. However, these promoters have limitations, including growth dependence and metabolic sensitivity. The limited number of available heterologous promoters, including the thiamine promoter *PthiA* [60], the xylanase promoter PxylP [61], and the human estrogen receptor *hERα* [62], often suffer from significant limitations, including the requirement for specific growth conditions, leaky expression, or weak activity [60,62]. The synTFs responsive to small-molecule inducers offer a promising alternative, enabling tunable, orthogonal, and condition-independent gene regulation. For example, a tetracycline (Tet)-responsive synTF system has been developed for *Aspergillus fumigatus* [63] and *Aspergillus niger* [64], based on Tet repressor DNA binding domains (TetR DBD) combined with viral activation domains (VP16 AD) [65]. Vogt et al. enhanced the efficiency and sensitivity of this system for fungal applications by using improved transactivators (e.g., rtTA2S-M2) and optimized promoters containing multiple *tetO* operator sites. Moreover, the coupled Bm3R1 DBD and VP64 AD were implemented to establish synTF targeting the Bm3R1 binding site, located multiple copies upstream of the *Cp* minimal promoter for *A. niger*, *Trichoderma reesei* [66], and *Aspergillus oryzae* [67]. Rekdal et al. engineered *A. oryzae* mycelium using Bm3R1-based synTF to control multi-gene expression [67]. By tuning metabolic pathways, they were able to enhance the production of vitamins and amino acids in the mycelium while reducing off-flavors, resulting in a protein-rich, palatable fungal biomass that can serve as a next-generation food product. Meyer et al. employed a Tet-on synTF system in *A. niger* for inducible and tunable expression of biomass-degrading enzymes. Using inducible synTF enabled on-demand enzyme production for lignocellulose breakdown [66]. However, there are a few examples that highlight the need for a broader synTF toolbox, which is currently lacking. A larger synTF library enables greater flexibility in designing complex genetic circuits, facilitates the independent regulation of multiple genes, and is essential for applications such as multi-gene metabolic engineering, synthetic pathway optimization, and dynamic control of multi-protein assemblies. To meet these demands, a wide range of other DBDs can be explored, including transcription activator-like effectors (TALEs), CRISPR/dCas systems (such as dCas9 or dCas12), and DBDs derived from heterologous TFs. Notably, higher plants possess over 2000 TFs across more than 60 families (e.g., PlantTFDB), with nearly half classified as plant-specific. These represent a rich and largely untapped source of orthogonal DBDs suitable for synthetic biology in fungi [68]. By placing such heterologous TFs downstream of small-molecule-responsive promoters, such as *tetO* (tetracycline-inducible) or *lacO* (IPTG-inducible), it is possible to engineer synTFs whose activity can be externally controlled. Furthermore, using a single synTF to target a bidirectional promoter containing a synTF binding site to activate the expression of two enzymes was recently reported by Rekdal et al. [67] by fusing two copies of the Bm3R1 binding site with two minimal promoters, *gpdA* and *hhfA,* to drive dual mCherry and GFP expression in *A. oryzae*. Building on such designs, the coordinated use of synTFs to regulate multi-gene pathways represents a powerful strategy for engineering complex biosynthetic routes by enabling the efficient reuse of well-characterized synTFs. Advances in regulation reprogramming tools allow for the expansion of modularity and inducibility of gene expression systems in filamentous fungi, while minimizing the impact on cell growth and fitness.

### 3.2. Genome Editing Tools

Engineering can involve gene deletion, gene addition, the formation of mixed pathways, the engineering of scaffold synthases, and the design of tailored enzymes. Precise genome editing and targeted DNA delivery are essential for engineering microbial cell factories utilizing synthetic genetic circuits, particularly in non-domesticated filamentous fungi, as efficiency may be compromised by the non-homologous end-joining (NHEJ) repair pathway. This occurs because NHEJ repairs DNA breaks by directly ligating the broken ends without requiring homologous templates, often resulting in the random integration of donor DNA. Enhancing homologous recombination can be achieved through the deletion of *ku70*, *ku80*, and *lig4* genes [69,70] or by utilizing long homologous sequences [71], which can improve the accuracy and efficiency of integration [72]. However, homologous recombination-based genetic manipulation in filamentous fungi remains less efficient than in *E. coli* and *S. cerevisiae*. The powerful genome editing based on type II clustered regularly interspaced short palindromic repeats (CRISPR)-associated protein, primarily Cas9 and Cas12a, is now applied to filamentous fungi. This technology accelerates genetic modification, enabling more precise and efficient genome editing. The modular cloning toolkit developed by Mozsik et al. [71] consists of non-integrative fungal shuttle vectors, such as those harboring the AMA1 sequence [73], which enable autonomous replication in filamentous fungi, including *A. niger* and *Trichoderma reesei*. This toolkit also enables efficient delivery of CRISPR components, including Cas9 and sgRNA, into cells for genome integration without requiring long homologous sequences and selection markers. Rekdal et al. developed an RNP-based CRISPR-Cas9 method tailored for filamentous fungi, i.e., *A. oryzae* [67]. This approach uses ribonucleoprotein complexes (RNPs) assembled from commercially available Cas9 protein and sgRNAs, bypassing the need for plasmids or Cas9 expression. The DNA repair template, harboring a pyrG selectable marker flanked by short homologous arms, enables precise integration. A phenotypic screening system ensures locus-specific insertion by allowing only correctly integrated constructs to loop out the pyrG marker, surviving in media with 5-fluoroorotic acid. This system significantly improves efficiency, minimizes off-target effects, and supports sequential marker recycling, making it a robust alternative to traditional homologous recombination and plasmid-based delivery of the CRISPR system. The CRISPR/Cas tool has thus far been successfully implemented in filamentous fungi to introduce BGCs into the genome [74,75]. Recently, Fang et al. engineered *T. reesei* using in vivo multi-fragment assembly combined with CRISPR/Cas9-mediated integration [76]. In a single transformation step, they successfully assembled, integrated, and expressed a 32.7 kb sorbicillinoid BGC, the largest BGC reported to date, directly assembled in filamentous fungi, to improve cellulase gene regulation [76]. Neutral loci, identified through computational analysis and experimental validation, allow consistent, high-level gene expression in *A. oryzae* without affecting growth or morphology, as demonstrated in the work by Rekdal et al. [67]. Additionally, the synthetic orthogonal region can serve as a versatile integration site, facilitating precise and highly efficient integration of non-native DNAs, thereby advancing synthetic biology applications and enhancing systematic genetic engineering in filamentous fungi. While the majority of genome editing tools, including those discussed above, have been developed and optimized in Ascomycetes, such as *Aspergillus* and *Trichoderma*, it is essential to note that similar strategies are now being applied to Basidiomycetes, discussed in recent reviews by Zhang et al. (2023) and Jain et al. (2024) [77,78]. For example, the successful use of CRISPR/Cas9-mediated gene editing in *Coprinopsis cinerea* has established this Basidiomycete as a tractable genetic model for functional genomics [79]. Similarly, the application of gene-editing strategies in the *Ustilago maydis* has enabled targeted manipulation of biosynthetic pathways [80]. Application of CRISPR/Cas9-mediated gene editing in *P. ostreatus* has confirmed the feasibility of applying these approaches in filamentous Basidiomycetes of industrial importance [81]. Together, these studies illustrate how tools developed in Ascomycetes are providing a foundation for, and increasingly being translated to, the genetic engineering of Basidiomycetes, hopefully paving the way for broader applications in filamentous members.

### 3.3. Optimization Methods for Heterologous Gene Expression

Advances in biotechnology have introduced various strategies to enhance protein production and metabolic output of filamentous fungi [31]. These include codon optimization, substituting native signal peptides with more effective alternatives [82], incorporating carrier proteins [83], and modifying glycosylation patterns [84,85]. Additional approaches include refining cellular stress responses, such as the unfolded protein response and endoplasmic reticulum-associated degradation [86,87]. Additionally, these approaches include implementing feedback circuits to balance competing biosynthetic demands [71], enhancing intracellular transport mechanisms [88], facilitating unconventional secretion pathways [88], and designing strains with reduced protease activity [89].

Here, we elaborate on the transformative potential of combinatorial optimization, a powerful strategy that has not been considered elsewhere. Indeed, a significant concern is that achieving efficient material production in filamentous fungi is challenging due to unknown optimal expression levels of various genes and potential bottlenecks caused by the unbalanced expression of heterologous proteins. Moreover, to improve material properties produced by filamentous fungi, structural proteins might need to be released into the cell environment and then attached to the outer parts of fungal cells. To achieve this, other enzymes are required to work together to assemble the extracellular matrix proteins into functional units. As a result, multiple constructs need to be constructed and tested, with lessons learned from unsuccessful attempts informing the development of a construct that produces the desired output. This trial-and-error process can be extremely time-consuming. The complex physiology of filamentous fungi, slow growth, and the inherent challenges of genetic transformation make the traditional design–build–test–learn (DBTL) cycle significantly slower and less predictable than in model organisms like *E. coli* or *S. cerevisiae.* Combinatorial optimization provides a solution by enabling the rapid generation of numerous genetic constructs [90]. By simultaneously combining different gene variants and promoters, this approach enables the creation of a large number of genetic constructs, ranging from thousands to millions. As a result, multiple strains are created using the combinatorial library of generated genetic constructs. Subsequent screening of these strains allows for the identification of one that exhibits optimal gene expression and metabolite production. Naseri et al. previously established a combinatorial optimization method, so-called COMPASS, for rapid expression-balancing of metabolic pathway genes in yeast [91]. However, to our knowledge, such a combinatorial optimization methodology has not been reported for filamentous fungi. Although initially developed for yeast, this method has strong potential for adaptation to filamentous fungi, as its core principle, systematically combining gene variants and orthogonal regulatory elements to balance expression, can be directly applied to generate combinatorial libraries in these organisms. As shown in Figure 2, the combinatorial optimization workflow involves assembling diverse combinations of promoters, genes, and terminators into transcriptional units, which are then combined into multi-gene constructs and integrated into the fungal genome (see also Section 3.2). This parallelized strategy allows researchers to bypass time-consuming sequential engineering and instead screen large libraries for high-performing strains.

Although the combinatorial assembly strategy is a promising technology, its implementation in filamentous fungi faces some challenges that limit its efficiency. Screening top performers within a combinatorial library of thousands to millions of variants is time-consuming due to the slow growth and complex cultivation requirements of filamentous fungi, making high-throughput screening difficult and time-consuming. Additionally, the multinucleate nature and complex regulatory networks in filamentous fungi further reduce predictability because interactions between synthetic heterologous constructs and endogenous pathways may lead to heterogeneous populations and unpredictable outcomes. Technical challenges in multi-gene construct assembly, coupled with long cultivation times and resource-intensive screening, increase the total cost and complexity. Therefore, the practical application of combinatorial optimization needs careful consideration of these biological, technical, and logistical limitations. Fortunately, ongoing advances in automation, high-throughput screening technologies, and biosensors hold great promise to overcome these obstacles, paving the way for more efficient and predictable engineering of filamentous fungi for sustainable material production.

## 4. Fabrication of Fungal Materials

While synthetic biology enables precise control over gene expression and material-related traits, realizing the full potential of these engineered filamentous fungi requires optimized fabrication processes, which are critical to producing high-quality, application-specific fungal materials at scale. Therefore, in this section, we focus on the practical methods used to cultivate and process fungal materials, ranging from solid-state and liquid-state fermentation to advanced AM.

Fungal materials can be derived from fruiting bodies [20,92] and mycelium. Mycelium can be used pure or as a composite, incorporating other materials (e.g., agricultural waste, natural and/or synthetic fibers or substances), making it often more versatile, rigid, and durable. Another common way to classify mycelium materials is by the cultivation conditions, including (1) solid-state fermentation (SSF), (2) liquid-state fermentation (LSF), and (3) additive manufacturing approaches. Each method can be adapted with variations, and SSF and LSF can be combined in specific applications (Table 2) [11,18,29,37,49]. The production process involves multiple steps: selecting, mechanically processing/homogenizing, and hydrating lignocellulosic substrates (e.g., sawdust, straw, and wood chips); sterilization; inoculation with fungal cultures; growth under controlled conditions; and drying/curing or finishing (Figure 3). [12,27]. Substrate water content typically ranges between 50% and 70%, modeled after commercial mushroom production [93]. Alternatively, liquid media may serve as a nutrient base in static or bioreactor setups [9,11,48]. To prevent contamination, substrates are sterilized, commonly using steam. Inoculation is performed with spores, fungal tissue, or colonized grains (spawn), followed by growth in shape-defining compartments [13,27,49] such as containers, trays, or molds under controlled conditions (25–32 °C [29], 60–100% [29] relative humidity) commonly for days, weeks, or months, depending on the intended outcome [12,27,94]. AM provides a pathway with substrate-core deposition, wood filament scaffolds, and bio-inks that combine substrate and fungi. Bio-inks are pre-mixed with fungal species, while the other methods require an additional inoculation step [29,37].

Commercial production of mycelium-based composites was pioneered by Evocative [95,96,97]. Their SSF-based techniques involve growing mycelium in molds to produce 3D shapes or flexible mats [9,11]. To create mats, the top fungal layer is harvested from substrates separated by woven/non-woven fabrics or metallic meshes to avoid chunks of lignocellulosic material in the final product [98,99]. The separation layer can also be made of particles [100] of different sizes than the nutrient base. Additionally, the separation layers can also serve as scaffolds, promoting 3D growth for flexible composites [98,99]. Conditions such as elevated temperatures (~30 °C), high humidity (99%), and increased CO_2_ (50–70 k ppm) stimulate aerial hyphal growth, resulting in stronger, more robust networks [98,99]. These conditions promote growth away from the nutrient base along a CO_2_ gradient established by fungal metabolism, allowing the organism to escape the high concentration near the surface [11]. Stimulation or damaging the mycelium while growing can also increase the strength and robustness of the network [13,97]. Alternatively, mats can be cultivated on liquid or gel-like media in static trays or bioreactors before being cured into shape [9,11,49,101]. Mats produced using those methods are typically more fragile than those from solid substrates [11,49]. Alternatively, a pre-colonized solid lignocellulosic material can be blended with water or an aqueous solution to create a gel-like slurry, which is then poured into a static container for incubation with a fungal mat growing on the surface of the slurry [102]. Overall, this process, while promising, remains challenging due to its dependence on skilled personnel and the scalability of growth phases [29].

**Table 2 jof-11-00721-t002:** Different production techniques to fabricate pure and composite fungal materials.

Production Method	Description	Advantages	Disadvantages
Solid-State Fermentation (SSF) [9,11,95,96,98,99]	Fungal growth occurs on a solid substrate with minimal moisture (30–80%), utilizing pre-colonized lignocellulosic substrates, such as sawdust, in molds.	- Economically competitive with LSF - High-volume productivity - Low water, energy, and sterility requirements (pasteurization can be an option) - Reduced contamination risk - Cheap substrates	- Prolonged colonization times - Variability in growth rates - Scalability challenges - Variability, depending on nutrient profile
Liquid-State Fermentation (LSF) [9,11,49,101]	(1) Submerged (bioreactor): Involves growth of fungi in a liquid medium, resulting in a microbial suspension.	- Fast growth rates - Lower space requirements, and easy harvesting (vs. SSF) - Closed system - Good distribution of nutrients and O_2_ - Good scalability and automation potential	- Sensitive to contamination - Requires complete sterilization - Higher valued media - Fragile fungal mass - High capital costs for automated systems/monitoring
	(2) Liquid State Surface Fermentation (LSSF): Static systems using shallow trays; Involves the growth of fungi in a liquid medium with high moisture content, resulting in an amicrobial suspension.	- Fast growth rates - Good scalability - Lower space requirements, and easy harvesting (vs. SSF) - Simple and lower cost (vs. bioreactor) - Flexible manipulation while incubation	- Sensitive to contamination - Requires complete sterilization - Higher valued media - Fragile fungal mass - Limited size - Longer incubation in bioreactors - Possible O_2_ and nutrient gradients
Additive Manufacturing [37] (AM), 3D printing	(1) Substrate Core Deposition: mostly lignocellulosic paste-like material is deposited layer-by-layer.	- Customizable scaffolds - Precise control over structure and design, rapid prototyping - Potential for automation	- Limited to small-scale applications - Material consistency can vary
	(2) Filament-Based Scaffolds: Utilize filaments containing nutrients for mycelium bonding, printed first and inoculated afterward.	- As above - Promotes mycelium growth	- Requires specific filaments - May not be widely available - Limited substrate choice
	(3) Bio-Inks: Integrates organic substrates (carrier) and living fungal cells into an extrudable paste.	- Allows for combined fabrication and inoculation - Carrier also binds non-lignocellulosic material - Can produce load-bearing structures - Potential for automation	- Complex formulation - Performance varies based on the used materials - Higher contamination risk
Solid-Substrate Surface Fermentation (SSSF) [102,103]	Filamentous fungi grow on the surface of a liquid medium, utilizing nutrients from a submerged solid substrate, and form high-density mats.	- High-density mats - Minimal waste - Rapid cell growth - Low energy use	- Requires careful control of pH and carbon-to-nitrogen ratios - Oxygenation challenges - Higher contamination risk as SSF
Living Fungal Slurry [49,102]	Blending a pre-colonized solid medium with an aqueous solution creates a gel-like slurry, resulting in a homogeneous distribution of hyphae when poured into trays.	- Lower contamination risk as LSF - Consistent properties (vs. SSF) - Simplified harvesting process (vs. SSF)	- Requires precise control of blending and incubation conditions - Complex initial setup, additional growth time, and higher contamination risk as SSF

## 5. Challenges and Future Directions

Fungal materials based on filamentous species represent an innovative approach to material production, leveraging the natural growth of fungi to create sustainable alternatives to synthetic and conventional materials due to their renewable nature and biodegradability [1,9,18]. To make a positive impact on our lives, particular challenges need to be addressed. Filamentous fungi have emerged as a significant platform for material production due to their remarkable structural capabilities, inherent pre-mRNA splicing system, and access to abundant biosynthetic precursors and coenzymes. Fungal materials outperform conventional materials in certain areas. Still, due to their often foam-like mechanical properties, high water absorption, natural variations, susceptibility to wear and abrasion forces, biodegradability, and gaps in material property documentation and compliance with regulatory bodies, their applications are usually limited to specific applications and non-load-bearing structures [27,29,37].

Another significant hurdle remains the slow process of manufacturing [27] and the scalability, along with the rather complex, multi-step processes of the cultivation of fungal materials, which ultimately decrease their market competitiveness [29,37]. While using cultivation methods such as SSF and LSF offers advantages in terms of productivity and growth rates, they also present inherent limitations. For instance, SSF presents several challenges, notably the necessity for accurate regulation of operational parameters such as temperature (heat transfer), humidity, substrate composition, moisture, and inoculation, all of which directly affect the yield of fungal growth on bulk substrates. Furthermore, it is often vital to maintain optimal CO_2_ concentrations when using certain fungi to encourage mycelium growth while inhibiting fruiting body formation. Effective contamination management is crucial (also in LSSF) because of the extensive operational spaces involved [29]. Techniques like AM allow for customizable scaffolds and precise control over structure, but they are often limited to small-scale applications and can suffer from material inconsistencies [37]. Automation of fungal cultivation processes is another area where challenges persist. Many existing production techniques are labor-intensive and require multiple steps and careful monitoring of parameters such as temperature, humidity, and nutrient concentrations. As a result, the integration of automated systems and smart algorithms for large-scale production remains a critical yet underexplored avenue for advancing fungal material cultivation [11,29,37]. To address these challenges, the authors—representing a smaller research group—have successfully developed a semi-automated system for the production of mycelial materials, which has been disclosed (WO2025131307A1) [104]. The disclosure highlights a well-defined technical contribution aimed at improving cultivation efficiency and reducing labor demands through advanced process control and monitoring. Intellectual property (IP) can support innovation when applied to clearly original developments. However, one major factor complicating the landscape for fungal materials is the IP environment, particularly in areas that involve common knowledge within the mycology community, which has been dominated by a few key players. The patents available in this domain often raise concerns by covering broadly known methodologies, thereby limiting access to essential tools for researchers and smaller enterprises, ultimately slowing progress in applications such as packaging, construction, and textiles [105,106].

When evaluating the economic feasibility of fungal-based materials, it is essential to acknowledge that, although these products offer sustainable benefits, their production processes are currently more complex and time-consuming compared to traditional oil-based alternatives. This complexity can affect their cost-effectiveness and market competitiveness. Furthermore, although fungal materials are biodegradable and environmentally friendly, their durability may limit their application in demanding environments. Given that these materials are a new class, it is essential to explore tailored regulatory frameworks that consider their unique characteristics, ensuring compliance while fostering innovation in this emerging field.

No current filamentous fungal chassis serves as a universal platform for heterologous pathway reconstitution toward improving their capacity for material production [107]. This limitation stems from their complex biology, which includes multiple nuclei, a robust cell wall that complicates transformation, and poorly understood mechanisms of gene expression, regulation, and horizontal gene transfer [107]. These complexities often lead to unpredictable outcomes in genetic modifications, causing variations or failures in enhancing specific traits. Notably, recent advancements in the emerging field of synthetic biology have emerged as a promising solution, enabling the engineering of these fungi as chassis organisms. These innovations have enabled the development of more orthogonal tools and methods that minimally interfere with the host’s endogenous regulatory mechanisms, allowing for the robust, orthogonal, and tunable production of enzymes of interest. To address the genetic engineering challenges, we highlighted here advanced synthetic biology tools and methodologies as potential solutions for optimizing filamentous fungal chassis to effectively express the target proteins and enzymes. For instance, inducible synTFs offer dynamic control over the expression of both heterologous and endogenous genes. Refining CRISPR/Cas toolkits can facilitate multi-gene manipulations and streamline processes for integrating large synthetic genetic circuits into fungal genomes. Traditional biosynthetic research often relies on the classical design–build–test–learn (DBTL) cycle, which can be labor-intensive, particularly for filamentous fungi with poorly understood biological systems, particularly during the “building” stage of developing high-performance fungal strains. To mitigate this, we propose the application of combinatorial optimization methodologies to accelerate the building step of DBTL cycles. Once such tools and methods are developed, standardization and centralization efforts are also critical for broader implementation. This progress holds great promise for the future of genetic engineering in filamentous fungi.

However, it should be emphasized that chemical engineering laboratories need to adopt standard design principles, such as the BioBrick rules [108], to design toolboxes and contribute parts to repositories like the BioBrick repository or the Galaxy-SynBioCADa portal [109]. Publicly accessible datasets generated through combinatorial optimization should be uploaded to third-party facilities like CloudLabs, biofoundries, and the Synthetic Biology Foundry. Centralizing these datasets would enable the development of deep learning algorithms to design rational metabolic engineering strategies [110], further advancing material production capabilities in filamentous fungi. By streamlining and accelerating the DBTL cycle through these strategies, research groups worldwide can enhance the efficiency and scalability of bioengineering projects focused on filamentous fungi [52,111].

Fungal research is also hindered by further significant obstacles, including insufficient infrastructure for handling extensive datasets that are often unsearchable. Moreover, discrepancies in the quality of genome annotations create challenges for data comparison and interpretation. To address these challenges, future efforts should focus on developing robust infrastructure, securing sustainable funding models, and standardizing genome annotation processes to enhance data accessibility and reliability in the field [1,112].

## 6. Conclusions

Fungi play a pivotal role in human and planetary health. In this paper, we briefly explored diverse applications of fungi, with a particular focus on materials and their potential across various industries. The examination of both rigid and flexible fungal material cultivation highlights their adaptability, environmental benefits, and innovative potential across various industries. Furthermore, the role of genetic engineering in enhancing the properties of these materials was emphasized, showcasing successful modifications that improve resilience and material characteristics while also addressing the challenges faced in this domain. The analysis of fabrication techniques highlighted the significant influence of cultivation methods on material properties and customization potential. However, fungal materials encounter challenges related to scalability, automation, reproducibility, and intellectual property, as well as the need for improved genetic manipulation to fully realize their potential.

Looking ahead, the field of fungal materials is poised for transformative growth. Advancements in synthetic biology and bioengineering promise to overcome current limitations, enabling the creation of highly specialized materials tailored for applications ranging from construction to biomedical fields. The integration of artificial intelligence (AI) and automation into fungal cultivation processes has the potential to revolutionize production efficiency and scalability. AI-driven algorithms can optimize growth conditions in real-time by analyzing data from environmental sensors, thereby ensuring that parameters such as temperature, humidity, and nutrient concentrations are maintained at optimal levels. This precision can significantly reduce the labor intensity and variability associated with traditional cultivation methods. Furthermore, machine learning models can predict growth patterns and yield outcomes based on historical data, enabling proactive adjustments to cultivation strategies. Automating processes such as substrate preparation, inoculation, and monitoring can enhance reproducibility and consistency in material properties, making fungal products more competitive with conventional alternatives. By leveraging AI and automation, the industry can not only streamline production but also facilitate the development of innovative applications for fungal materials across various sectors. Moreover, collaborative efforts across academic disciplines such as synthetic biology and bioprocessing, alongside non-academic stakeholders including industry and regulatory bodies, will be essential to develop robust frameworks for scaling up production while ensuring quality and sustainability.

As the global demand for sustainable solutions grows, fungal materials have the potential to redefine industries and lead the transition to a circular economy. With strategic investment in research, innovation, and infrastructure, fungi can become a cornerstone of future sustainable technologies, addressing not only environmental challenges but also enhancing human quality of life. The next decade could see fungi move from niche applications to mainstream adoption, fundamentally reshaping our approach to materials and resource management.

## Figures and Tables

**Figure 1 jof-11-00721-f001:**
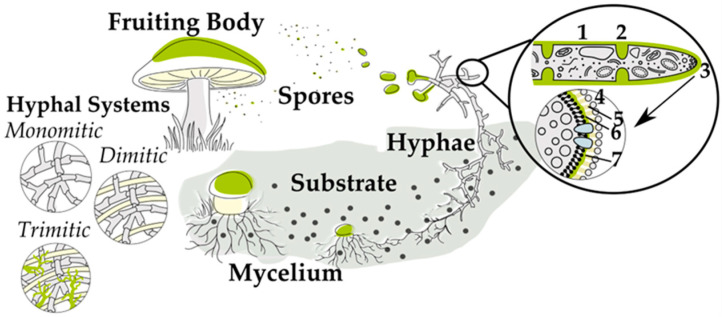
Life cycle of Basidiomycetes. There are four key stages, including spore germination, hyphal growth, mycelium development, and fruiting body formation. Hyphal interconnections occur in three forms: monomitic, dimitic, and trimitic. The zoomed-in section provides a detailed view of the hyphal structure, showcasing the cell wall (1), septum (2), growth tip with Spitzenkörper (3), mannoproteins (4), β-glucans (5), chitin (6), and the plasma membrane (7).

**Figure 2 jof-11-00721-f002:**
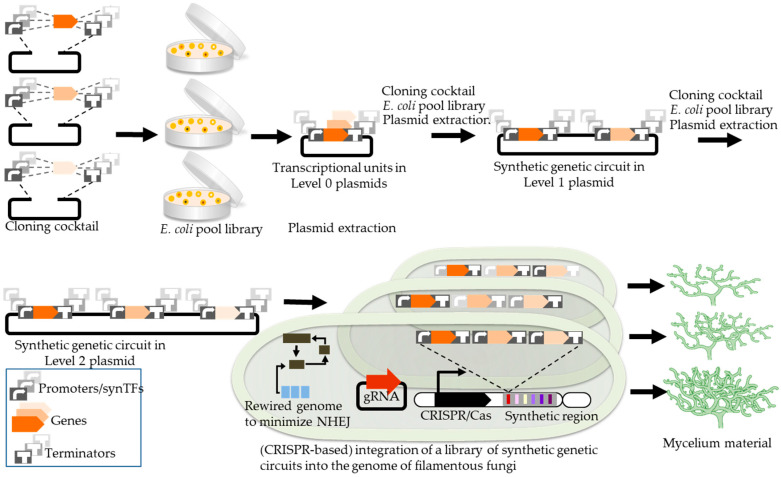
Workflow for building combinatorial libraries in filamentous fungi. The creation of combinatorial libraries in filamentous fungi follows a stepwise process of assembling, amplifying, and integrating genetic components into a host genome. It starts with a one-pot reaction mixture containing libraries of genetic elements, such as promoters or synthetic transcription factors (synTFs), genes, and terminators. These elements are combined in diverse arrangements during a single cloning step, using either homology-based cloning or classical enzyme-assisted digestion/ligation methods in *Escherichia coli* (*E. coli*) serving as a cloning host. Each fragment in the library is engineered for seamless assembly: promoters and terminators are designed to match the boundaries of insertion sites in Level 0 plasmids. The downstream region of a promoter overlaps with the upstream sequence of its associated gene, while the downstream region of the gene aligns with the upstream region of its corresponding terminator. The initial step creates transcriptional units for each gene (e.g., three genes are illustrated here), which are then combined in subsequent reactions to form multi-gene constructs in Level 1 plasmids. Subsequently, a plasmid library containing all possible gene combinations is created (e.g., three genes in Level 2 plasmids as shown here), and the finalized plasmids are prepared for stable genome integration. This method introduces linearized plasmids into specific genomic regions through clustered regularly interspaced palindromic repeats (CRISPR/Cas) and single-guide RNA (sgRNA) targeting the unique region in successive transformation steps.

**Figure 3 jof-11-00721-f003:**
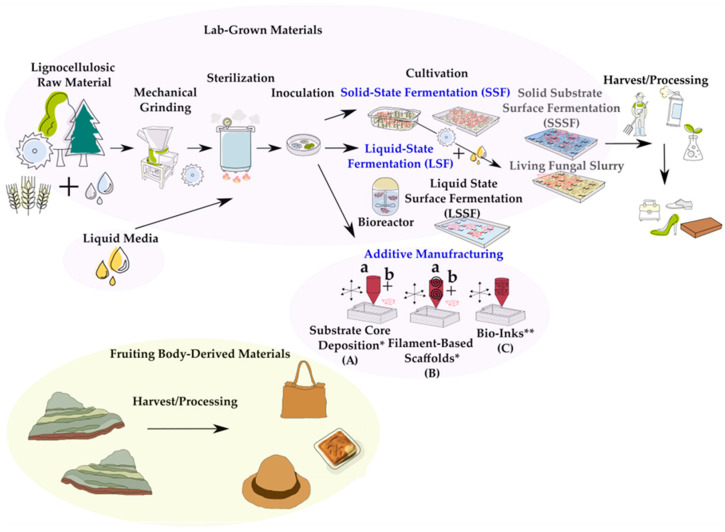
Manufacturing of fungal materials derived from fruiting bodies or lab-grown mycelium. Processes include solid-state fermentation (SSF) using container and tray setups, liquid-state fermentation (LSF) (submerged in a bioreactor), and liquid-state surface fermentation (LSSF). Mixed methods include solid-substrate surface fermentation (SSSF), where mycelium grows on the surface of a liquid medium using nutrients from a submerged solid substrate, and living fungal slurry, created by blending pre-colonized solid medium with an aqueous solution. Additive manufacturing approaches include (A) substrate core deposition (paste), (B) filament-based scaffolds, and (C) bio-inks. * Inoculum is added after substrate deposition. ** Substrate and fungi are applied together.

**Table 1 jof-11-00721-t001:** Performance characteristics of mycelium composites.

Property	Range
Compressive Strength	0.17 to 1.1 MPa
Tensile Strength	0.03 to 0.18 MPa
Density	59 to 552 kg/m^3^
Flexural Strength	0.05 to 0.29 MPa
Acoustic Absorbance	70% to 75% at 1000 Hz
Moisture Uptake	40 to 580 wt%
Thermal Conductivity	0.04 to 0.18 W/mK
Fire Resistance	Varies by composition

## Data Availability

Not applicable.

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
