# Peer review of "Fungal Innovations—Advancing Sustainable Materials, Genetics, and Applications for Industry"

_jof, 2025, doi:10.3390/jof11100721_

Round 1
Reviewer 1 Report
Fungi play a pivotal role in human and planetary health. This current manuscript provides a brief overview of various applications of fungi, with a particular focus on both rigid and flexible fungal materials. The obtained results could be useful information for researchers who worked in the similar field. In general, this is an interesting review.
Some suggestions are as follows:
- Section 3. Synthetic Biology Toolbox for Genetic Engineering of Fungal Materials. The author should streamline the descriptions of the methods and add some examples of using these methods to prepare fungal materials.
- Line 52, 106, 348, 352. Error! Reference source not found. The author should check the references.
Author Response
Major comments
Fungi play a pivotal role in human and planetary health. This current manuscript provides a brief overview
of various applications of fungi, with a particular focus on both rigid and flexible fungal materials. The
obtained results could be useful information for researchers who worked in the similar field. In general, this
is an interesting review.
Detailed comments
Some suggestions are as follows:
1. Section 3. Synthetic Biology Toolbox for Genetic Engineering of Fungal Materials. The author should
streamline the descriptions of the methods and add some examples of using these methods to prepare
fungal materials.
RESPONSE:
In response to this comment, we have added examples to the section. All changes are highlighted in blue.
For details, please see: Page 6, lines 225–233; Page 7, lines 286–290; and Page 8, lines 329–334.
2. Line 52, 106, 348, 352. Error! Reference source not found. The author should check the references.
RESPONSE:
We have reviewed and corrected the references accordingly.
Reviewer 2 Report
The current manuscript focuses on fungal materials, their development, and the use of synthetic biology for their production. While this is a relevant and timely topic, frequently discussed in sustainability-focused literature, the manuscript presents several issues that need to be addressed.
The opening sentences of the abstract read more like a general introduction rather than a concise summary of the manuscript. Consider reducing the contextual background to better highlight the core contributions of your work.
Although the English is generally well written, there are instances of awkward phrasing and unclear sentences. For example, line 63: “decomposers of the forest” and line 79: “lion share” do not sound scientific. These and similar expressions should be revised for clarity and tone.
Please ensure the manuscript is thoroughly proofread before submission. Phrases such as “(Error! Reference source not found.)” should not appear in the final version.
Overall, the manuscript would benefit from a more comprehensive and structured presentation of information. I hope these comments are helpful for your revision.
In general, the figures are difficult to interpret and may not be essential to the manuscript. Notably, Figures 1 and 3 are not referenced in the text. If these figures are not clearly contextualised or directly relevant, I recommend removing them. If you choose to keep Figure 1, consider increasing its size to improve clarity. Figure 2 should be simplified for better readability. For Figure 3, if all acronyms are explained in the legend, it may be more visually effective to include them in the figure itself.
And again, if the figures do not add substantial value, consider deleting them. Moreover, if any figures are adapted from other sources, include the proper references.
Lines 60 and 130: Please ensure that scientific names are written correctly throughout the manuscript.
Lines 78–80: You mention several genera; within those, are there specific species that are preferred or more advantageous? If so, please add information on why and how.
Lines 80 and 193: Review punctuation here and in the entire manuscript.
Line 105: You refer to fungal strains; should fungal species also be mentioned? Clarify whether both are relevant in this context.
Lines 116–127: This section would benefit from visual aids, such as photos, schemes, or even a table to present the information more clearly.
Lines 160–165: There are many other examples that could be included here.
In Section 3, it would be helpful to include examples, or even case studies, of how synthetic biology techniques are currently being applied in fungal product production. You mention challenges but do not discuss potential solutions; this would be a valuable addition to the discussion.
Line 219: Maintain consistency in formatting. For example, “e.g.,” is italicised in some places but not in others. Choose one style and apply it throughout the manuscript.
Line 226: Check spacing throughout the document, not just here.
Line 330: A comparative analysis of "fabrication techniques" would largely improve this section.
There is some repetition in the discussion of fermentation methods (e.g., SSF and LSF). Please revise to avoid repetition of information here and everywhere else.
Line 408: Scalability is mentioned, but economic feasibility is not addressed. Are fungal-based commercial products more cost-effective than traditional alternatives? How long do they last? Are they biodegradable or environmentally impactful? You mention patents, but is there any existing legislation regulating these products?
Lines 495–496: Finally, the mention of AI and automation in cultivation is very interesting, but it needs more details. I suggest expanding this point and adding on discussion about it.
Author Response
Major comments
The current manuscript focuses on fungal materials, their development, and the use of synthetic biology
for their production. While this is a relevant and timely topic, frequently discussed in sustainability-focused
literature, the manuscript presents several issues that need to be addressed.
The opening sentences of the abstract read more like a general introduction rather than a concise
summary of the manuscript. Consider reducing the contextual background to better highlight the core
contributions of your work.
Although the English is generally well written, there are instances of awkward phrasing and unclear
sentences. For example, line 63: “decomposers of the forest” and line 79: “lion share” do not sound
scientific. These and similar expressions should be revised for clarity and tone. Please ensure the
manuscript is thoroughly proofread before submission. Phrases such as “(Error! Reference source not
found.)” should not appear in the final version.
RESPONSE:
The expressions have been replaced with appropriate terms: “decomposers of the forest” was changed to
“wood-decaying fungi” and “lion’s share” to “majority.” In addition, in response to this comment, we carefully
reviewed the manuscript and replaced other inappropriate or informal expressions throughout the text. The
text was proofread, and the errors in the references were corrected. All changes have been highlighted in
blue. All changes were highlighted in blue.
Overall, the manuscript would benefit from a more comprehensive and structured presentation of
information. I hope these comments are helpful for your revision.
Detailed comments
In general, the figures are difficult to interpret and may not be essential to the manuscript. Notably, Figures
1 and 3 are not referenced in the text. If these figures are not clearly contextualised or directly relevant, I
recommend removing them.
RESPONSE:
The figure legends have been rephrased to improve clarity and ease of understanding. During the final
submission, a formatting error disrupted the cross-referencing in the draft; these references have now been
corrected. The manuscript has also been revised to better contextualize the figures. All changes have been
highlighted in blue.
If you choose to keep Figure 1, consider increasing its size to improve clarity.
RESPONSE:
The size has been increased.
Figure 2 should be simplified for better readability.
RESPONSE:
The figure has been simplified as requested by the reviewer.
For Figure 3, if all acronyms are explained in the legend, it may be more visually effective to include them
in the figure itself. And again, if the figures do not add substantial value, consider deleting them. Moreover,
if any figures are adapted from other sources, include the proper references.
RESPONSE:
Acronyms have been included in the figure, and the figure has been improved for better visibility. Moreover,
figure legends have been rephrased. The figure was created originally, though it may include common
elements that help illustrate the subject matter. If any adaptations from other sources are present, proper
references are provided.
Lines 60 and 130: Please ensure that scientific names are written correctly throughout the manuscript.
RESPONSE:
The species name has been corrected (“Fomes Fomentarius” to “Fomes fomentarius” and for Piptoporus
betulinus, a more recent name has also been included (Fomitopsis betulina).
Lines 78–80: You mention several genera; within those, are there specific species that are preferred or more
advantageous? If so, please add information on why and how.
RESPONSE:
In response to the reviewers' comments, we acknowledge the importance of specifying which species within
the mentioned genera are preferred for mycelium-based applications (page 1, lines 39-41; page 2, lines 81-
91).
Moreover, we modified the text and added another reference (page 2, line 91).
Additionally, in Section 2 (page 3, lines 117-121), the importance of the hyphal type was highlighted.
Lines 80 and 193:
RESPONSE:
Review punctuation here and in the entire manuscript. Punctuation has been corrected and carefully
reviewed.
Line 105: You refer to fungal strains; should fungal species also be mentioned? Clarify whether both are
relevant in this context.
RESPONSE:
The comment has been addressed by clarifying that species influences are more pronounced, particularly
regarding growth (page 3, line 115).”
Lines 116–127: This section would benefit from visual aids, such as photos, schemes, or even a table to
present the information more clearly.
RESPONSE:
In response to the reviewer’s comment, we have revised the section to include a more straightforward
presentation of the architectural projects by listing them with their respective materials and completion years.
Lines 160–165: There are many other examples that could be included here.
RESPONSE:
Thank you for your valuable feedback regarding the inclusion of additional examples. While we acknowledge
that there are many other companies in this field, we have chosen to focus on well-known names to provide
clarity and maintain a concise presentation. Our intent was not to exclude others but rather to highlight
significant examples that are widely recognized. If you have specific names or examples that you believe
should be included, we would greatly appreciate your suggestions, as they could further enhance the paper.
In Section 3, it would be helpful to include examples, or even case studies, of how synthetic biology
techniques are currently being applied in fungal product production. You mention challenges but do not
discuss potential solutions; this would be a valuable addition to the discussion.
RESPONSE:
In response to this comment, we have added examples to the section. For details, please see: Page 6,
lines 225–233; Page 7, lines 286–290; and Page 8, lines 329–334.
We also thank the reviewer for pointing out the need to offer potential solutions to the challenges. The
discussion of possible solutions to genetic engineering challenges in filamentous fungi was included in the
original manuscript. To better highlight these points, the introduction to the solutions has been highlighted
more clearly. In particular, we have referred to the tools and combinatorial optimization approaches,
standardization efforts, and infrastructure improvements (pages 13 – 14, lines 500 -514).
Line 219: Maintain consistency in formatting. For example, “e.g.,” is italicised in some places but not in
others. Choose one style and apply it throughout the manuscript.
RESPONSE:
Done.
Line 226: Check spacing throughout the document, not just here.
RESPONSE:
Done.
Line 330: A comparative analysis of "fabrication techniques" would largely improve this section.There is
some repetition in the discussion of fermentation methods (e.g., SSF and LSF). Please revise to avoid
repetition of information here and everywhere else.
RESPONSE:
Thank you for your constructive feedback regarding the comparative analysis of fabrication techniques. In
response, the table has been revised to better highlight the differences between the various methods while
minimizing repetition. The structure of the “Additive Manufacturing” section has been retained to maintain
clarity regarding the distinct techniques involved (all Table 1). We acknowledge that direct comparisons can
be challenging, as production methods often involve different materials, strains, substrates, and target
markets. We believe these revisions enhance the overall readability and effectiveness of the comparative
analysis.
Line 408: Scalability is mentioned, but economic feasibility is not addressed. Are fungal-based commercial
products more cost-effective than traditional alternatives? How long do they last? Are they biodegradable
or environmentally impactful? You mention patents, but is there any existing legislation regulating these
products?
RESPONSE:
Thank you for your insightful comments regarding the scalability and economic feasibility of fungal-based
materials. To address the mentioned concerns and we point out that the relevant information was already
provided in section 4 (Fabrication, and also Table 1), and section 5 (challenges), as well as in the
description about rigid and flexible materials (page 4, lines 130-132, 159-162, 166-169), which cover:
1. Economic Feasibility: While the text does not explicitly compare the cost-effectiveness of fungal-based
products to traditional alternatives, it highlights the challenges of scalability and the complexities
involved in the production processes. These factors can influence economic viability. Specifically, the
discussion on the slow manufacturing processes and multi-step cultivation methods (see sections
discussing SSF and LSF) indicates that these challenges can impact market competitiveness, which
indirectly addresses economic feasibility. The production process involves multiple steps, including
selecting and processing lignocellulosic substrates, sterilization, inoculation, and growth under
controlled conditions, which can take days, weeks, or even months. In contrast, oil-based products can
often be produced more quickly and efficiently, which may make them more economically attractive.
2. Durability: The text mentions that fungal materials often have foam-like mechanical properties and are
susceptible to wear and abrasion, which implies limitations in their longevity and durability for certain
applications. There are also certain mechanical testing results given. This suggests that while they may
be suitable for non-load-bearing structures, their lifespan could be a concern in more demanding
environments.
3. Biodegradability and Environmental Impact: The text states that fungal materials are biodegradable
and represent sustainable alternatives to synthetic materials, addressing their environmental impact
positively. This is covered in the opening statement (section 5) regarding their renewable nature and
biodegradability.
4. Legislation and Regulation: The mention of regulatory bodies and the need for compliance in material
property documentation suggests that there are existing legislative frameworks governing the use of
fungal-based products. However, specific details about current regulations are not extensively covered
in the text. It is important to note that fungal materials are a very new class of materials and are often
addressed under conventional material regulations, which may not fully account for their unique
properties and potential applications. The discussion about intellectual property (IP) challenges
indicates the competitive landscape and the implications of existing patents, which can relate to
regulatory environments.
We appreciate your feedback and added a suitable paragraph in section 5 (page #, lines # - #): “When
evaluating the economic feasibility of fungal-based materials, it is essential to acknowledge that,
although these products offer sustainable benefits, their production processes are currently more
complex and time-consuming compared to traditional oil-based alternatives. This complexity can affect
their cost-effectiveness and market competitiveness. Furthermore, although fungal materials are
biodegradable and environmentally friendly, their durability may limit their application in demanding
environments. Given that these materials are a new class, it is essential to explore tailored regulatory
frameworks that consider their unique characteristics, ensuring compliance while fostering innovation
in this emerging field.”
Lines 495–496: Finally, the mention of AI and automation in cultivation is very interesting, but it needs
more details. I suggest expanding this point and adding on discussion about it.
RESPONSE:
Thank you for your suggestion regarding AI and automation in fungal cultivation. In the current manuscript,
these topics are briefly addressed in Section 5 and, in response to this comment, further discussed in
Section 6 and Table 1 to highlight their potential for improving efficiency, scalability, and reproducibility of
mycelium production. However, the AI and automation are not the primary focus of this review, and the
current literature on these topics remains fragmented across environmental control, predictive modeling,
and automated processing. A comprehensive discussion would need a dedicated review to combine
methodologies, case studies, and technological trends. Therefore, we believe a separate perspective would
be necessary to cover this area.
Section 6 (pages 14 -15, lines 548 - 561): “The integration of artificial intelligence (AI) and automation into
fungal cultivation processes has the potential to revolutionize production efficiency and scalability. AI-driven
algorithms can optimize growth conditions in real-time by analyzing data from environmental sensors,
thereby ensuring that parameters such as temperature, humidity, and nutrient concentrations are
maintained at optimal levels. This precision can significantly reduce the labor intensity and variability
associated with traditional cultivation methods. Furthermore, machine learning models can predict growth
patterns and yield outcomes based on historical data, enabling proactive adjustments to cultivation
strategies. Automating processes such as substrate preparation, inoculation, and monitoring can enhance
reproducibility and consistency in material properties, making fungal products more competitive with
conventional alternatives. By leveraging AI and automation, the industry can not only streamline production
but also facilitate the development of innovative applications for fungal materials across various sectors”.
Reviewer 3 Report
Dear authors, I have reviewed your review paper proposal. Overall, the proposal is innovative and of interest to the scientific community; 70% of the bibliography is from the last 10 years. However, before considering its publication, there are some conceptual gaps that need to be clarified.
The first gap is the very definition of Fungal; the fungal kingdom encompasses a wide variety of microorganisms that, although they belong to the same kingdom, exhibit significant differences. For example, filamentous fungi, yeasts, mushrooms, mold, mycorrhizae, etc. Certainly, the review does not cover all types of fungi. So they should be specific and provide a clear definition of what they mean by fungi. Likewise, throughout the text they mention topics or concepts that are not explored in depth or are irrelevant, such as E. coli or artificial intelligence, which are briefly mentioned but not actually linked to the review.
The authors should also clarify what type of review they conducted—narrative or systematic—and, if systematic, indicate the methodology they followed, including the inclusion and exclusion criteria. In Section 2, they provide an extensive review of examples of rigid and flexible materials, but they do not discuss in depth how the fungal structure influences these physical properties. Could the composition of chitin, β-glucans, or other chemical compounds, as well as their spatial arrangement, have something to do with it?
From line 108 onward, they provide some important parameter values generated by composites; you should clarify the composite’s composition and discuss it to understand why those values are obtained. You should review your bibliography and avoid mixing different types of fungi, as their composition and behavior are very different—for example, yeast and filamentous fungi.
The sections that cover molecular biology and genetics mostly focus on the development of specific bioactive compounds, but they don’t address the fungi used to produce materials rigid and flexible.
Section 3.3 should discuss challenges and disadvantages.
Section 4 should go with Section 2.
On line 337, they mention fostering interaction between bioengineering and bioprocesses, but the review is deficient in its discussion of bioprocess engineering; unit operations should be included. Likewise, it is suggested to include a table of free and non-free patents and discuss it.
The authors’ proposal may be publishable, but they should focus their review to avoid confusion and conceptual gaps.
Some observations:
Lines 23 and 337: there is a contradiction in the stated goals; they are presenting three different goals. (aims or goal?
Line 41: Authors must define the fungal subkingdom or phyla under study.
Please review the style, grammar, and spelling throughout the document; for example, lines 276 and 278, line 282, line 363,line 425 and others.
Author Response
Dear authors, I have reviewed your review paper proposal. Overall, the proposal is innovative and of interest
to the scientific community; 70% of the bibliography is from the last 10 years. However, before considering
its publication, there are some conceptual gaps that need to be clarified.
The first gap is the very definition of Fungal; the fungal kingdom encompasses a wide variety of
microorganisms that, although they belong to the same kingdom, exhibit significant differences. For
example, filamentous fungi, yeasts, mushrooms, mold, mycorrhizae, etc. Certainly, the review does not
cover all types of fungi. So they should be specific and provide a clear definition of what they mean by fungi.
RESPONSE:
Thank you for your comment regarding the definition of fungi. We have clarified in the “Introduction” section
that this review focuses explicitly on filamentous fungi, particularly genera such as Pleurotus, Ganoderma,
and Trametes, which are central to the production of mycelium-based composites. All changes were
highlighted in blue.
Likewise, throughout the text they mention topics or concepts that are not explored in depth or are
irrelevant, such as E. coli or artificial intelligence, which are briefly mentioned but not actually linked to the
review.
RESPONSE:
Regarding E. coli, an appropriate change has been made to section 3 (page 5, lines 185 - 190) to clarify
why it is mentioned in this article.
Regarding AI, we already provided information on this topic and automation in Section 5. Additionally, we
have included further details in Section 6 (pages 14-15, lines 548-561), as well as in Table 1 (and also in
response to the last comment from Reviewer 2). While we briefly touch on this topic, it is not the primary
focus of the review; however, these enhancements emphasize the potential of AI and automation to improve
the efficiency and scalability of mycelium production.
The authors should also clarify what type of review they conducted—narrative or systematic—and, if
systematic, indicate the methodology they followed, including the inclusion and exclusion criteria.
RESPONSE:
This work is a narrative review. Our goal was not to conduct a systematic literature review but to provide an
overview of recent advances in fungal materials, with a special focus on synthetic biology and fabrication
approaches. We created findings from key publications to illustrate current developments, highlight
emerging opportunities, and discuss challenges, rather than applying formal systematic review methodology
with predefined inclusion and exclusion criteria. In response to this comment, changes were added to the
end of the introduction (page 3, lines 93 -97).
In Section 2, they provide an extensive review of examples of rigid and flexible materials, but they do not
discuss in depth how the fungal structure influences these physical properties. Could the composition of
chitin, β-glucans, or other chemical compounds, as well as their spatial arrangement, have something to
do with it?
RESPONSE:
In response to this and other comments, we acknowledge the importance of specifying which species within
the mentioned genera are preferred for mycelium-based applications; the text has been modified, and
references regarding the importance of the cell wall composition and hyphae type have been added (page
2, lines 80-91; page 3, lines 116 -120).
Furthermore, on page 3, lines 118-120, the importance of the hyphae type is highlighted.
From line 108 onward, they provide some important parameter values generated by composites; you should
clarify the composite’s composition and discuss it to understand why those values are obtained. You should
review your bibliography and avoid mixing different types of fungi, as their composition and behavior are
very different—for example, yeast and filamentous fungi.
RESPONSE:
Thank you for your valuable feedback regarding the parameter values obtained from rigid fungal
composites. To clarify, these composites are created by combining filamentous fungi with lignocellulosic
fibers or particles, with properties that vary based on the finishing method (e.g., hot/cold pressing), the
substrate used, and the specific fungal species and strains. It is essential to note that yeast, although a
crucial organism for genetics and the basic understanding of fungi, does not play a role in the mycelium
material section, as it is a single-celled organism that does not form networks like filamentous fungi.
The mechanical properties of these composites can fluctuate due to their biological nature and the influence
of various factors, including the type of hyphal system present. For instance, Ganoderma has a trimitic
hyphal system that enhances strength through the directional alignment of skeletal hyphae. However,
comparing the mechanical properties of different composites can be challenging due to variations in strains,
cultivation conditions, and substrates. This variability is why we present ranges for the properties—such as
compressive strength (0.17 to 1.1 MPa), tensile strength (0.03 to 0.18 MPa), and others—rather than
specific values. A comprehensive overview of these material properties can be found in articles by Sharma
et al., Aiduang et al., and Jones et al.
However, to address this concern, we added information on page 2, lines 82-91.
The sections that cover molecular biology and genetics mostly focus on the development of specific
bioactive compounds, but they don’t address the fungi used to produce materials rigid and flexible.
RESPONSE:
The authors thank the reviewer for this insightful comment. While much work has focused on fungi as
producers of bioactive compounds, filamentous fungi for rigid and flexible material production remain largely
underexplored from a genetic engineering perspective. Our review aims to highlight this gap by discussing
how advances in synthetic biology and combinatorial optimization can be explicitly applied to filamentous
fungi for material applications. We believe this focus complements existing literature and is essential for
advancing sustainable fungal materials.
Section 3.3 should discuss challenges and disadvantages.
RESPONSE:
We thank Reviewer 3 for the comment. In response, a discussion of the challenges and disadvantages of
combinatorial assembly has been added at the end of Section 3.3 (page 8, lines 340–354), highlighting that
slow growth, complex cultivation, multinucleate cells, and regulatory complexity hinder high-throughput
screening and increase cost and effort.
Section 4 should go with Section 2.
RESPONSE:
While Section 2 provides an overview of fungal materials and their properties, we have intentionally placed
Section 4 on fabrication methods after the discussion of genetic engineering (Section 3) for clarity of the
story. By first presenting the synthetic biology tools and strategies used to engineer filamentous fungi, the
subsequent section on fabrication logically builds on this basis, highlighting how engineered strains can be
translated into practical material production. This structure emphasizes the story from genetic design to
large-scale production of fungal materials.
On line 337, they mention fostering interaction between bioengineering and bioprocesses, but the review
is deficient in its discussion of bioprocess engineering; unit operations should be included. Likewise, it is
suggested to include a table of free and non-free patents and discuss it. The authors’ proposal may be
publishable, but they should focus their review to avoid confusion and conceptual gaps.
RESPONSE:
The authors see the reviewer’s point regarding the inclusion of a table on free and non-free patents.
However, we believe that providing such a table may not be feasible or appropriate for this review. Indeed,
the patent landscape for fungal materials is complex and rapidly evolving, with many patents covering broad
methodologies that may not accurately reflect current practices or access conditions. Instead of a detailed
patent analysis, we have referenced relevant reviews that thoroughly examine the patent situation in this
field, such as the works by Elsacker et al. (2023) and Meyer & Mengel (2024). These references offer
valuable insights into the current patent landscape, helping to interpret the implications of existing patents
on the development of mycelium-based materials. By focusing on these comprehensive reviews, we aim to
encourage innovation while acknowledging the challenges posed by the intellectual property environment.
Detailed comments
Some observations:
Lines 23 and 337: there is a contradiction in the stated goals; they are presenting three different goals.
(aims or goal?
RESPONSE:
Thank you for your comment. The Abstract has been rephrased to eliminate the contradiction and to
present the consistent goal of the manuscript (page 1, lines 17–24).
Line 41: Authors must define the fungal subkingdom or phyla under study.
RESPONSE:
Thank you for your comment. In response to this comment and another comment of Reviewer 3, we have
clarified in the “Introduction” section (page 1, lines 39 - 41) that this review specifically focuses on
filamentous fungi, particularly genera such as Pleurotus, Ganoderma, and Trametes, which are central to
the production of mycelium-based composites.
Please review the style, grammar, and spelling throughout the document; for example, lines 276 and 278,
line 282, line 363,line 425 and others.
RESPONSE:
We have thoroughly revised the manuscript to correct issues of style, grammar, and spelling, including those
indicated, with all changes highlighted in blue.